# Nutrient Patterns and Its Association and Metabolic Syndrome among Chinese Children and Adolescents Aged 7–17

**DOI:** 10.3390/nu15010117

**Published:** 2022-12-27

**Authors:** Jia Shi, Hongyun Fang, Xue Cheng, Qiya Guo, Lahong Ju, Wei Piao, Xiaoli Xu, Dongmei Yu, Liyun Zhao, Li He

**Affiliations:** 1National Institute for Nutrition and Health, Chinese Center for Disease Control and Prevention, Beijing 100050, China; 2NHC Key Laboratory of Trace Element Nutrition, National Institute for Nutrition and Health, Chinese Center for Disease Control and Prevention, Beijing 100050, China

**Keywords:** nutrient pattern, metabolic syndrome, children and adolescents, China

## Abstract

This study was designed to explore the associations between nutrient patterns (NPs) and metabolic syndrome (MetS) and its five components among Chinese children and adolescents aged 7–17. The required data of participants were collected from the China National Nutrition and Health Surveillance of Children and Lactating Mothers in 2016–2017. Ultimately, 13,071 participants were included. Nutrient patterns were obtained by means of factor analysis. Multivariate logistic regression analysis was conducted to evaluate the association between nutrient patterns with MetS and its components. After adjusting covariates, the results of logistic regression models revealed that high-carbohydrate patterns were associated with the presence of abdominal obesity. The high-animal protein pattern was negatively associated with high triglyceride (TG) and low high-density lipoprotein cholesterol (HDL-C). The high-sodium-and-fat pattern had a negative relationship with elevated blood pressure (BP) and was positively associated with low HDL-C. The high-Vitamin D-and-Vitamin B_12_ pattern had protective effects on MetS, high TG, and low HDL-C. Further large-scale longitudinal investigations are necessary in the future.

## 1. Introduction

Metabolic syndrome (MetS) is characterized by cardiovascular risk factors including abdominal obesity, elevated blood pressure (BP), dyslipidemia (high triglyceride (TG) and low high-density lipoprotein cholesterol (HDL-C)), and elevated fast blood glucose (FBG) [1]. The prevalence of MetS has been increasing, and has now become a main public health issue in numerous countries [2]. According to a previous study, MetS prevalence among adults varies from 20% to 27% in developing countries and is even higher in developed countries [3]. However, less attention has been placed on MetS in children and adolescents, compared it in adults. Actually, the increasing trend in the prevalence of MetS is obvious among young populations in China. A meta-analysis demonstrated that the prevalence of MetS increased from 2.3% in 2004–2010 to 3.2% in 2011–2014 [4]. A recent study reported that this number had elevated to 5.98% in 2016–2017, which has almost doubled over the past few years [5]. The onset of MetS in those of a young age could lead to MetS in adulthood, resulting in cardiovascular disease and type 2 diabetes mellitus (T2DM) in later life [6,7]. Therefore, it is worthwhile to identify risk factors of MetS and prevent them as early as possible.

A complicated combination of genetic, metabolic, and environmental factors, including ethnic predisposition, physical activity, smoking, diet, and so on, contribute to the etiology of MetS and its components [2,8]. Undoubtedly, as a major lifestyle factor, diet has been proved to be strongly associated with MetS [9]. With that said, dietary intervention is considered as an effective strategy for the prevention and treatment of MetS [10,11]. In addition to ample evidence indicating an inverse association between Vitamin B_12_, Vitamin C, and Vitamin D with MetS [12,13,14], a high intake of dietary fiber, whole grains, fruits, and vegetables have been reported to have independent protective effects against the incidence of MetS [15,16,17]. While these studies have mainly focused on individual nutrients and food, little attention has been placed on the patterns of dietary intake, especially nutrient patterns.

It has previously been reported that dietary patterns may provide valuable perception of diet associated disease compared to single food or nutrients, which may be more predictive of the risk of chronic disease [18,19]. Dietary pattern approaches comprise data driven methods such as factor analysis, which allows the dietary information to determine the unique dietary pattern for the population to be evaluated [20]. Therefore, we have previously published a study about dietary patterns and MetS [21], based upon food groups in the same study population as that presented here. However, there are diverse foods with different cooking and preservation methods all over the world, therefore, dietary patterns based on food cannot reasonably explain the underlying mechanisms of disease [19]. Nutrient patterns (NPs), a combination of multiple nutrients, may provide more realistic data on the possible biological mechanisms linking diet and disease, particularly when comparing dietary intakes across different nations [8,22]. Since nutrients are components of diet played a role in human health, and are universal in the world, they cannot be affected by diverse cooking and food preservation methods [19].

Existing limited studies, investigating the combination of nutrients, have focused on cancer [23], bone mineral density [24], obesity [25,26], cognitive health [27], and metabolic syndrome [2,8,12,28] among adults. Until present, there has been no study evaluating the association between nutrient patterns and MetS and its components among the young age group. Therefore, the objective of this study is to identify nutrient patterns and examine the relationship between NPs with MetS and its components among Chinese children and adolescents aged 7–17.

## 2. Materials and Methods

### 2.1. Sampling Method and Participants

Data were from the China National Nutrition and Health Surveillance of Children and Lactating Mothers in 2016–2017. The details of the study design, inclusion and exclusion criteria were all previously reported [29]. In total, 13,071 participants aged 7–17 were included in this study, with informed consent obtained from all subjects before the start of the investigation.

### 2.2. Anthropometric Measurements and Clinical Examinations

The anthropometric measurement indicators, including height, waist circumference (WC), systolic blood pressure (SBP), and diastolic blood pressure (DBP), were conducted by trained investigators. Clinical examinations included the blood concentration of triglycerides (TG), fast blood glucose (FBG), and high-density lipoprotein cholesterol (HDL-C), details of which can be found in our published paper [21].

### 2.3. The Definition of MetS

According to Cook’s criteria [30], if 3 of the 5 criteria below are met, MetS can be diagnosed. 

(1) Abdominal obesity: WC ≥ age-and-sex-specific 90th percentile, obtained from the cutoff points for Chinese children and adolescents [31];

(2) High triglycerides: serum TG ≥ 1.24 mmol/L;

(3) Low HDL-C: HDL-C ≤ 1.03 mmol/L;

(4) Elevated blood pressure: SBP or DBP ≥ age-sex-and-height 90th percentile [32];

(5) Elevated fast blood glucose: FBG ≥ 6.1 mmol/L.

### 2.4. Dietary and Nutrient Assessment

Detailed descriptions of dietary measurements are provided elsewhere [21]. In brief, the diet data was obtained by means of a 24-h dietary recall for 3 days. The Chinese Food Composition Tables [33,34] were used to analyze the food consumption data (grams per day per reference man) and to determine the intake levels of energy and nutrients. A total of 14 food groups were included for partial correlation with NPs in our samples, including cereals, tubers, mixed beans, legumes, vegetables, fruits, nuts, meat and poultry, fish and shrimp, milk, eggs, oil (vegetable oil and animal oil), salt, fast foods, ethnic foods, and cakes.

### 2.5. Nutrient Patterns

In the present study, we used 25 types of nutrients to identify nutrient patterns, including protein, carbohydrate, fat, cholesterol, dietary fiber, Vitamin A, Vitamin D, Vitamin E, Vitamin B_1_, Vitamin B_2_, niacin, Vitamin B_6_, folate, Vitamin B_12_, Vitamin C, calcium, phosphorus, potassium, magnesium, sodium, iron, zinc, selenium, copper, and manganese. Using principal component analysis (PCA) with VARIMAX rotation, factor analysis was conducted to extract nutrient patterns. All the necessary prerequisites of PCA including the Kaiser–Meyer–Olkin (KMO) measure of 0.922 and the significant Bartlett’s test of sphericity (*p* < 0.001) were met. The number of extracted factors was decided by eigenvalues greater than 1, the scree plot, and the interpretability [35]. NPs were named by nutrient variables whose absolute factor loading was over 0.300. Factor scores for each NP were calculated by summing up intakes of nutrients weighted by their factor loadings. The largest score showed that the diet of participants tended to correspond with the NP. Given that simple linear dose-response relationships are unlikely to be found in nutritional epidemiology [25], we categorized the participants based on quartiles of factor scores for each NP.

### 2.6. Other Covariates

Data of sociodemographic and lifestyle variables were collected using a standardized questionnaire. Among those, the classifications of residence, age group, and Engel’s coefficient were consistent with our previous study [21]. Physical activity was grouped as low level (0–3 days/week), high level (≥4 days/week), and none [36]. Both the passive smoking and alcohol drinking status were divided into 2 groups (yes and no). Screen time was described as “<2 h” and “≥2 h” [37]. Family size was categorized as “≤3”, “=4”, “=5”, and “>5”.

### 2.7. Statistical Analysis

SAS 9.4 (SAS Institute Inc., Cary, NC, USA), SPSS 26 (SPSS Inc., Chicago, IL, USA), and R 4.1.2 were used. The categorical data was collected as numbers (percentage) and the Chi-square test was conducted. Given that the data was not normally distributed, the continuous variable was reported as the median (Inter Quartile Range, IQR), and the Kruskal–Wallis test was used for statistical analysis. The correlation coefficients between NPs and food groups adjusted for sex, age, residence area, and energy intake were calculated using the partial correlation test. The associations between the quartiles of NPs and the odds of MetS and its components were assessed using multivariate logistic regression analysis, adjusted for potential confounding variables. The difference was statistically significant when the *p* value was <0.05.

## 3. Results

### 3.1. Nutrient Patterns and Its Correlation with Food Groups

#### 3.1.1. Nutrient Patterns

Five mutually exclusive NPs were identified (Figure 1), which accounted for 62.011% of the total variation. A total of 5 NPs were able to explain 25.223%, 18.863%, 7.494%, 5.275%, and 5.156% of the variance. NP1 was characterized by high factor loadings from carbohydrate, dietary fiber, protein, B-vitamins (Vitamin B_1_, Vitamin B_2_, niacin, and folate), and a variety of minerals (magnesium, phosphorus, calcium, potassium, iron, zinc, selenium, copper, and manganese). NP2 was rich in protein, cholesterol, B-vitamins (Vitamin B_1_, Vitamin B_2_, niacin, and folate), minerals (magnesium, phosphorus, potassium, zinc, selenium, and calcium), Vitamin A, and fat. NP3 was greatly loaded with fat, sodium, and Vitamin E. NP4 was characterized by the high consumption of Vitamin A, Vitamin C, Vitamin B_2_, folate, and calcium. NP5 had a positive correlation with Vitamin B_12_, Vitamin D, Vitamin B_6_, and calcium. 

#### 3.1.2. Partial Correlation Test

The result of the correlation between NPs with food groups is shown in Table 1. NP1 had a moderately positive correlation with cereals, a weak positive correlation with tubers, mixed beans, legumes, vegetables, fruits, and nuts, a moderately negative correlation with oil, and a weak negative correlation with meat and poultry, eggs, and salt. NP2 indicated a moderately positive correlation with meat and poultry, fish and shrimp, and eggs, a weak positive correlation with mixed beans, legumes, vegetables, fruits, and nuts, and a weak negative correlation with cereals, tubers, oil, and salt. A significantly positive correlation with NP3 and oil, a moderately positive correlation with salt, and a moderately negative correlation with cereals were observed. NP4 was shown to have a weak positive relationship with vegetables, fruits, eggs, tubers, mixed beans, legumes, nuts, fish and shrimp, milk, oil, and salt, and a weak negative relationship with cereals, meat and poultry, fast foods, cakes, and ethnic foods. NP5 showed a weak positive association with fast foods, ethnic foods, and cakes, and fish and shrimp, and a weak negative association with cereals, tubers, meat and poultry, eggs, oil, and salt. 

After combining the results of the factor analysis with the partial correlation test, the five extracted NPs were finally named as high-carbohydrate pattern, high-animal protein pattern, high-sodium-and-fat pattern, high-Vitamin A-and-Vitamin C pattern and high Vitamin D-and-Vitamin B_12_ pattern in turn.

### 3.2. Basic Characteristics of Participants

#### 3.2.1. Basic Characteristics of Participants in 5 NPs

The distribution of five NPs are presented in Table 2. The percentages of NP1 to NP5 among children and adolescents aged 7–17 were 24.382%, 23.036%, 18.690%, 19.218%, and 14.674%, respectively. The NP1 (high-carbohydrate pattern) was the main nutrient pattern in all subjects (Table 2). The distribution of NPs was statistically different for sex, residence districts, age group, economic status, physical activity, alcohol drinking status, and family size. The total energy intake of participants in 5 NPs was statistically significant. Notably, NP1 appeared in the participants who were female, living in rural district, older, physically inactive, had large family sizes (≥4), and higher energy intake. Participants in NP2 (high-animal protein pattern) were more likely to be male, living in an urban district, younger, physically active, and have smaller family sizes (≤3).

#### 3.2.2. The Prevalence of MetS and Its Components in 5 NPs

As shown in Figure 2, elevated BP was the most prevalent among participants, and elevated FBG was the least prevalent disease. The prevalence of abdominal obesity, high TG, and low HDL-C were significantly different in the 5 NPs. Participants in NP2 (high-animal protein pattern) and NP5 (high-Vitamin D, B_12_ pattern) had a lower prevalence of high TG and low HDL-C. Participants in NP1(high-carbohydrate pattern) and NP2 were more likely to have a relatively high prevalence of abdominal obesity. Participants in NP3 (high-sodium, fat pattern) were more likely to have low HDL-C and a lower prevalence of abdominal obesity. Although not statistically significant, participants in NP3 and NP5 had a relatively low prevalence of elevated BP and MetS, respectively.

(The prevalence of abdominal obesity, high TG, and low HDL-C were significantly different among the 5 NPs.)

### 3.3. The Association between NPs and MetS and Its Components

The associations of NPs and MetS and its five components are demonstrated in Table 3. Compared with Q1, subjects in Q4 of NP1 had a positive effect on abdominal obesity (OR = 1.200, 95%CI: 1.018–1.416). NP2 was significantly associated with decreased odds of high TG (Q4 vs. Q1, OR = 0.850, 95%CI: 0.733–0.985, *p* for trend = 0.018) and low HDL-C (*p* for trend < 0.0001). NP3 was inversely related to elevated BP (Q3 vs. Q1, OR = 0.882, 95%CI: 0.796–0.977) and positively associated with low HDL-C (Q4 vs. Q1, OR = 1.222, 95%CI: 1.022–1.461), although the values of *p* were not statistically significant after adjusting for covariates. Negative associations were observed between NP5 and MetS (Q4 vs. Q1, OR = 0.782, 95%CI: 0.623–0.983), high TG (Q4 vs. Q1, OR = 0.839, 95%CI: 0.730–0.960), and low HDL-C (*p* for trend < 0.0001).

## 4. Discussion

We identified five nutrient patterns using factor analysis and established the correlation between 5 NPs with food groups by means of partial correlation. Although no associations of 5 NPs with elevated FBG and NP4 with MetS and its components were found, the associations of NPs with MetS and its components were observed.

Among the 5 NPs, NP1 (high-carbohydrate pattern) was mainly observed in participants, followed by NP2 (high-animal protein pattern), NP4 (high-Vitamin A-and-Vitamin C pattern), NP3 (high-sodium-and-fat pattern), and NP5 (high-Vitamin D-and-Vitamin B_12_ pattern), which implies that subjects had a predominantly plant-based diet. In our study, NP1 appeared in the participants who were female, living in rural areas, older, physically inactive, had larger family sizes (≥4), and higher energy intake. Participants assigned to NP2 were more likely to be male, living in an urban district, younger, physically active, and have a smaller family size (≤3). Furthermore, the prevalence of abdominal obesity, high TG, and low HDL-C were significantly different in the 5 NPs. Participants in NP2 (high-animal protein pattern) and NP5 (high-Vitamin D-and-Vitamin B_12_ pattern) had a lower prevalence of high TG and low HDL-C. Moreover, participants in NP1 and NP2 were more likely to have a relatively high prevalence of abdominal obesity. Furthermore, we found it interesting that participants assigned to NP3 were more likely to have low HDL-C.

NP1 (high-carbohydrate pattern), which was rich in carbohydrate, dietary fiber, protein, B-vitamins (Vitamin B_1_, Vitamin B_2_, niacin, and folate), and a variety of minerals (magnesium, phosphorus, calcium, potassium, iron, zinc, selenium, copper, and manganese), was plant-sourced, and had a positive relationship with cereals, legumes, and vegetables. This pattern was similar to results of previous studies [8,28,38], although the association between NP1 and MetS and its components was different. Mahdi Vajdi et al. found that adherence to the nutrient patterns rich in fiber, carbohydrate, Vitamin D, Vitamin B_6_, Vitamin C, Vitamin B_1_, Vitamin E, niacin, magnesium, potassium, linoleic acid, and docosahexaenoic acid (DHA) were related to a lower risk of MetS [2]. In the study conducted by Bian et al. [39], the “B-vitamins” pattern was inversely related to MetS in Chinese adults. In contrast, SS Khayyatzadeh et al. suggested that a pattern rich in copper, selenium, Vitamin A, Vitamin B_2_, and Vitamin B_12_ was related to a higher risk of MetS in women [12]. However, NP1 in our study did not demonstrate any associations with MetS, which could be explained by the fact that the interactions between nutrient and neutral MetS-inducing and MetS-protecting effects [3,40] resulted in a non-significant relationship. However, NP1 was found to have a positive association with abdominal obesity, and participants in this pattern had a higher energy intake. Although consumption of dietary fiber, Vitamin D, and minerals such as calcium, magnesium, and potassium has been proved to be inversely related to obesity [25,41,42,43,44], positive relationships have been found between carbohydrate and B-vitamins. In fact, it has been noted that a greater proportion of the energy intake was contributed by the carbohydrate intake in Asians [45]. A previous study suggested that carbohydrates, especially refined carbohydrates, have been shown to result in increased visceral adiposity, decreased insulin sensitivity, and the upregulation of hepatic de novo lipogenesis [46]. In addition, B-vitamins in this pattern, particularly Vitamin B_1_ [47] and niacin [48] have been positively associated with obesity. Because B-vitamins may stimulate appetite, long-term consumption may trigger excessive energy intake and weight gain [41]. Our finding on the association between NP1 and abdominal obesity supports our previous findings [21] on the link between dietary patterns and abdominal obesity.

NP2 (high-animal protein pattern) was characterized by high factor loadings of protein, cholesterol, B-vitamins (Vitamin B_1_, Vitamin B_2_, niacin, and folate), minerals (magnesium, phosphorus, potassium, zinc, selenium, and calcium), Vitamin A, and fat. After partial correlation testing, we found that NP2 had positive correlations with meat and poultry, fish and shrimp, and eggs and vegetables. Participants in NP2 seemingly had a higher prevalence of abdominal obesity: howver, no association was found using the multivariate logistic regression model. The combination of nutrients and obesity-inducing (Vitamin B_1_ [47], niacin [48], and fat [49]) and obesity-protecting (protein [50], magnesium [28], potassium [42], selenium [51], and folate [52]) effects in NP2 complicates our interpretation. Therefore, because of the interactions between these nutrients, the overall effect of this pattern resulted in a non-significant association between this pattern and abdominal obesity. We also found that participants in NP2 had a lower prevalence of high TG and low HDL-C than other NPs. Negative relationships were finally observed between NP2 with high TG and low HDL-C. Although this is not consistent with other studies, the associations can be explained. Khayyatzadeh et al. suggested that nutrient patterns with high selenium, Vitamin A, and B groups could decrease TG levels in men [12]. A previous study also found that magnesium and niacin could reduce oxidative stress and increase the concentration of HDL-C [53]. In addition, dietary cholesterol intake, which came mostly from eggs, was shown to significantly increase serum HDL-C [54], and the additional dietary cholesterol from eggs did not negatively affect serum lipids. Furthermore, calcium was also found to have beneficial effects on lipid profiles, and the relative mechanisms included formation of soaps with fatty acids and increased fecal fat excretion [55]. Although the high consumption of animal products such as meat, processed meat, and eggs, similar to Western patterns in Asian countries, is reported to increase the risk of diabetes, hypertriglyceridemia, cardiovascular disease, and obesity in young people [12], Asians still adhere to a plant-based diet, indicating the maintenance of a low-fat diet compared with Western countries.

NP3 (high-sodium-and-fat pattern) was rich in Vitamin E, fat, and sodium and had positive correlations with salt and oil. Participants in NP3 had a lower prevalence of abdominal obesity, but no association was observed between NP3 and abdominal obesity, after adjusting for confoundings. However, participants in NP3 were more likely to have low HDL-C. In addition, it was interesting to find that NP3 was negatively associated with elevated BP. In general, an excess of salt intake was known as a risk factor for the occurrence of hypertension [56,57]. After checking the original data, we found that the oil consumed by participants was mainly vegetable oil. Vegetable oil, unlike animal oil, contains abundant unsaturated fatty acids such as omega-3 fatty acids, which are thought to be associated with a relatively high concentration of HDL-C and low blood pressure [58]. However, there was not a definite conclusion regarding the relationship between unsaturated fatty acids and HDL-C. A study also reported that substituting saturated fatty acids with unsaturated fatty acids could decrease the level of HDL-C [59]. Further, Vitamin E, which is ingested with unsaturated fat-containing foods, has been reported to possess anti-inflammatory [60], anti-oxidative [61], and anti-hypercholesterolemic [62] properties via the regulation of various signal pathways [63]. Nevertheless, effects of Vitamin E on HDL-C level in human studies were contradictory, and data on Vitamin E in hypertensive patients were limited, with heterogeneous findings published in a review in 2017 [63]. Further researches are warranted to verify the relationships between these nutrients and diseases.

NP4(high-Vitamin A-and-Vitamin C pattern) was characterized by the high consumption of Vitamin A, Vitamin C, Vitamin B_2_, folate and calcium and was positively associated with vegetables and fruits. However, no correlation was observed between NP4 and MetS and its components. This pattern extracted was similar to one in a previous study [64], but the result of relationship between with the pattern and disease was different. Although Vitamin C from vegetables and fruits may potentially decrease the risk of non-communicable diseases [65], controversies still exist in the relationship between calcium, Vitamin A, Vitamin B_2_, and folate with MetS and its components [8,12]. In a word, these nutrients and their synergistic effect resulted in a non-significant association of NP4 with MetS and its components. 

NP5 (high-Vitamin D-and-Vitamin B_12_ pattern) was rich in Vitamin B_6_,Vitamin B_12_, Vitamin D, and calcium. Since this pattern had a weak positive association with fast foods, ethnic foods, and cakes, we regarded NP5 as a unique nutrient pattern among the young Chinese population. Given that there are numerous kinds of fast foods, ethnic foods, and cakes in China, it is readily understandable to observe such nutrient pattern. Participants in NP5 had a relatively low energy intake and protective effects of NP5 on MetS, high TG, and low HDL-C, which were eventually observed. Even though there were inconsistent conclusions between calcium and MetS [8], ample evidence shows inverse relationships between Vitamin B_12_, Vitamin B_6_, and Vitamin D with MetS [13,66,67,68,69], giving a statistically significant finding. Therefore, further investigations between calcium with MetS are needed to draw a definite conclusion.

Some caution is necessary when interpretating the results. First, this was a cross-sectional study. Therefore, it was difficult to determine causal associations. Additionally, although most risk factors were adjusted for, we could not rule out the possibility of unknown confounders. Moreover, the subjective decisions when conducting factor analysis, such as the choice of nutrients to be included, the number of patterns to be extracted, and preference of rotation method, should be taken into consideration while interpreting the results. Nevertheless, this method is useful for enhancing our understanding of the complex dietary factors implicated in chronic diseases [70]. Additional prospective studies are still required. The current study had some strengths, including the large sample size, scientific sampling method, and rigorous quality control measures. To the best of our knowledge, our study is the first to explore the association between nutrient patterns with MetS and its components in a young population.

## 5. Conclusions

In summary, five nutrient patterns were obtained among Chinese children and adolescents aged 7–17. After adjusting for confounding factors, the associations between NPs with MetS and its components were observed. The high-carbohydrate pattern had a positive association with abdominal obesity. The high-animal protein pattern was negatively associated with high TG and low HDL-C. The high-sodium-and-fat pattern had a negative association with elevated BP and had a positive relationship with low HDL-C. The high-Vitamin D-and-Vitamin B_12_ pattern was inversely related to MetS, high TG and low HDL-C. More studies are still warranted in the future.

## Figures and Tables

**Figure 1 nutrients-15-00117-f001:**
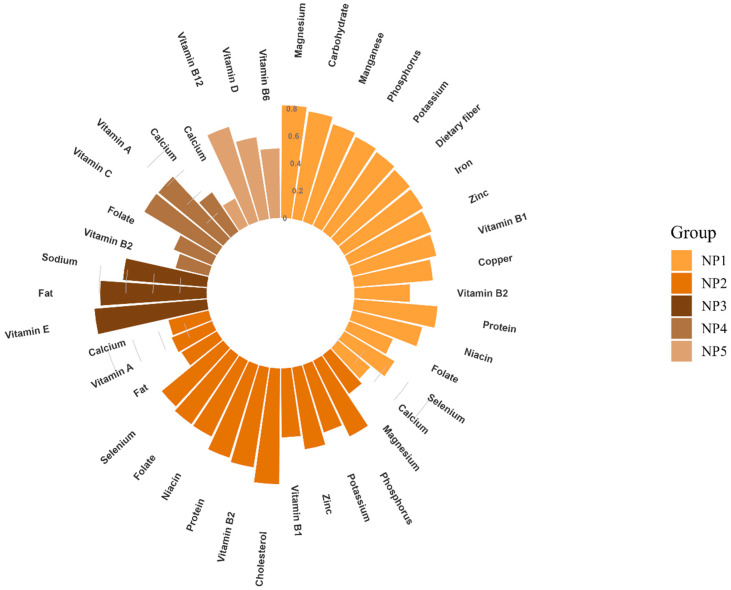
Factor loadings (all >0.300) of 5 Nutrient Patterns (NPs).

**Figure 2 nutrients-15-00117-f002:**
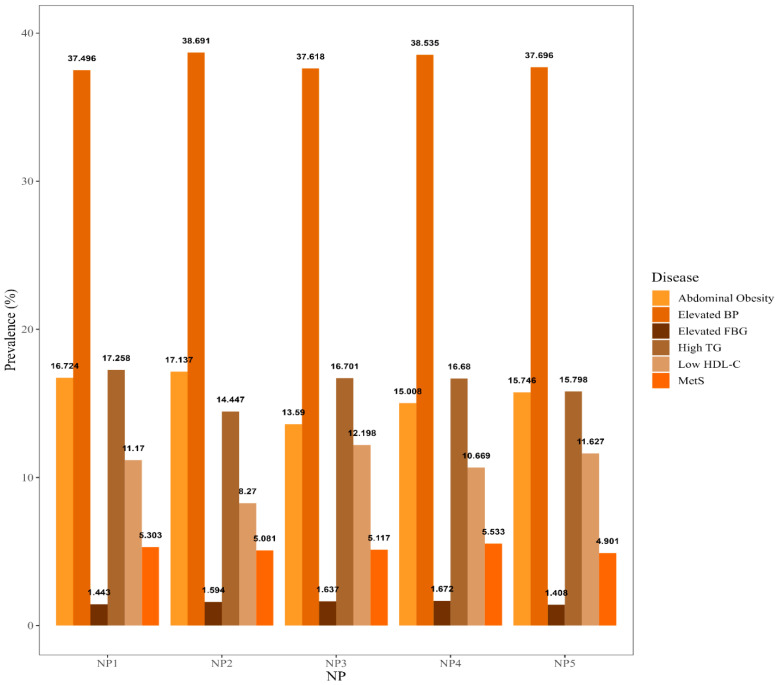
The prevalence of MetS and its components among 5 NPs.

**Table 1 nutrients-15-00117-t001:** The result of partial correlation test of nutrient patterns with food groups.

Food Group	NP1	NP2	NP3	NP4	NP5
Cereals	0.431 *	−0.183 *	−0.432 *	−0.088 *	−0.115 *
Tubers	0.172 *	−0.074 *	−0.039 *	0.075 *	−0.033 *
Mixed beans	0.096 *	0.023 *	−0.049 *	0.019 *	−0.005
Legumes	0.265 *	0.033 *	−0.082 *	0.071 *	−0.014
Vegetables	0.262 *	0.155 *	−0.112 *	0.285 *	−0.006
Fruits	0.120 *	0.088 *	−0.106 *	0.158 *	0.011
Nuts	0.071 *	0.049 *	−0.036 *	0.028 *	0.012
Meat and poultry	−0.054 *	0.517 *	−0.137 *	−0.065 *	−0.019 *
Fish and shrimp	0.036 *	0.345 *	−0.099 *	0.035 *	0.047 *
Milk	−0.003	0.013	−0.001	0.018 *	0.007
Eggs	−0.220 *	0.579 *	−0.092 *	0.135 *	−0.062 *
Fast foods, ethnic foods and cakes	−0.011	−0.004	−0.077 *	−0.071 *	0.236 *
Oil	−0.532 *	−0.242 *	0.739 *	0.026 *	−0.067 *
Salt	−0.084 *	−0.170 *	0.565 *	0.052 *	−0.057 *

Adjusted for sex, residence area, age and energy intake; * *p* < 0.05.

**Table 2 nutrients-15-00117-t002:** The basic characteristics [n (%)/Median (IQR)] of five NPs.

Characteristics	NP1	NP2	NP3	NP4	NP5	All
Sex *						
Male	1491(11.407)	1553(11.881)	1181(9.035)	1228(9.395)	1079(8.255)	6532(49.973)
Female	1696(12.975)	1458(11.154)	1262(9.655)	1284(9.823)	839(6.419)	6539(50.027)
Residence area *						
urban	1348(10.313)	1923(14.712)	817(6.250)	1103(8.439)	932(7.130)	6123(46.844)
rural	1839(14.069)	1088(8.324)	1626(12.440)	1409(10.780)	986(7.543)	6948(53.156)
Age group *						
prepubertal	1305(9.984)	1609(12.310)	1054(8.064)	1199(9.173)	784(5.998)	5951(45.528)
pubertal	953(7.291)	825(6.312)	710(5.432)	706(5.401)	506(3.871)	3700(28.307)
Post-pubertal	929(7.107)	577(4.414)	679(5.195)	607(4.644)	628(4.805)	3420(26.165)
Engel’s Coefficient *						
≥60%	64(0.490)	57(0.436)	33(0.252)	50(0.383)	16(0.122)	220(1.683)
50–59%	75(0.574)	56(0.428)	69(0.528)	55(0.421)	42(0.321)	297(2.272)
40–49%	66(0.505)	66(0.505)	65(0.497)	56(0.428)	50(0.383)	303(2.318)
30–39%	171(1.308)	168(1.285)	166(1.270)	145(1.109)	106(0.811)	756(5.784)
<30%	459(3.512)	440(3.366)	456(3.489)	331(2.532)	252(1.928)	1938(14.827)
unknown	2352(17.994)	2224(17.015)	1654(12.654)	1875(14.345)	1452(11.109)	9557(73.116)
Physical activity *						
none	764(5.845)	570(4.361)	578(4.422)	623(4.766)	448(3.427)	2983(22.822)
0–3 days/week	1078(8.247)	1085(8.301)	798(6.105)	916(7.008)	696(5.325)	4573(34.986)
≥4 days/week	1345(10.290)	1356(10.374)	1067(8.163)	973(7.444)	774(5.922)	5515(42.193)
Screen time						
<2 h	198(1.515)	193(1.477)	148(1.132)	156(1.193)	124(0.949)	819(6.266)
≥2 h	2989(22.867)	2818(21.559)	2295(17.558)	2356(18.025)	1794(13.725)	12,252(93.734)
Passive smoking						
yes	1400(10.711)	1266(9.686)	1110(8.492)	1074(8.217)	855(6.541)	5705(43.646)
no	1787(13.671)	1745(13.350)	1333(10.198)	1438(11.001)	1063(8.133)	7366(56.354)
Alcohol drinking *						
yes	462(3.535)	329(2.517)	368(2.815)	300(2.295)	296(2.265)	1755(13.427)
no	2725(20.848)	2682(20.519)	2075(15.875)	2212(16.923)	1622(12.409)	11,316(86.573)
Family size *						
≤3	709(5.424)	912(6.977)	516(3.948)	608(4.652)	548(4.192)	3293(25.193)
4	1217(9.311)	1011(7.735)	835(6.388)	846(6.472)	675(5.164)	4584(35.070)
5	636(4.866)	596(4.560)	559(4.277)	531(4.062)	372(2.846)	2694(20.611)
>5	625(4.782)	492(3.764)	533(4.078)	527(4.032)	323(2.471)	2500(19.126)
Energy intake *	2147.845(778.027)	1892.159(712.887)	2114.965(953.437)	1373.598(499.943)	1440.272(571.703)	1816.990(866.140)
**All**	3187(24.382)	3011(23.036)	2443(18.690)	2512(19.218)	1918(14.674)	13,071

Chi-square test and Kruskal–Wallis test were applied; * *p* < 0.0001; IQR: Inter Quartile Range.

**Table 3 nutrients-15-00117-t003:** The associations between nutrient patterns with MetS and its components.

NP	MetS	Abdominal Obesity	Elevated FBG	Elevated BP	High TG	Low HDL-C
	ORs (95%CIs)	ORs (95%CIs)	ORs (95%CIs)	ORs (95%CIs)	ORs (95%CIs)	ORs (95%CIs)
NP1						
Q1	1.000	1.000	1.000	1.000	1.000	1.000
Q2	1.078(0.860, 1.352)	0.994(0.865, 1.142)	0.905(0.609, 1.345)	0.952(0.860, 1.054)	0.982(0.857, 1.125)	0.942(0.803, 1.106)
Q3	1.233(0.975, 1.559)	1.099(0.951, 1.269)	1.072(0.718, 1.601)	1.027(0.923, 1.142)	1.061(0.921, 1.222)	1.068(0.904, 1.262)
Q4	1.207(0.917, 1.589)	**1.200(1.018, 1.416)**	0.708(0.430, 1.166)	0.954(0.843, 1.079)	1.133(0.963, 1.334)	1.043(0.857, 1.269)
*p* for trend	0.115	0.072	0.395	0.645	0.233	0.676
NP2						
Q1	1.000	1.000	1.000	1.000	1.000	1.000
Q2	1.056(0.849, 1.314)	1.045(0.909, 1.201)	0.839(0.556, 1.264)	1.035(0.936, 1.144)	0.905(0.794, 1.032)	**0.814(0.701, 0.946)**
Q3	0.946(0.753, 1.190)	1.040(0.902, 1.199)	1.077(0.725, 1.600)	1.008(0.908, 1.118)	0.874(0.763, 1.001)	**0.715(0.609, 0.838)**
Q4	0.882(0.688, 1.131)	1.073(0.923, 1.247)	0.920(0.592, 1.431)	1.056(0.944, 1.181)	**0.850(0.733, 0.985)**	**0.589(0.492, 0.706)**
*p* for trend	0.482	0.385	0.800	0.446	**0.018**	**<0.0001**
NP3						
Q1	1.000	1.000	1.000	1.000	1.000	1.000
Q2	1.013(0.818, 1.255)	0.949(0.830, 1.086)	0.824(0.558, 1.218)	0.973(0.880, 1.075)	0.983(0.861, 1.121)	0.995(0.850, 1.164)
Q3	0.924(0.739, 1.156)	1.000(0.873, 1.146)	0.819(0.548, 1.223)	**0.882(0.796, 0.977)**	0.935(0.816, 1.070)	0.888(0.752, 1.048)
Q4	0.919(0.713, 1.184)	0.942(0.808, 1.099)	0.968(0.626, 1.498)	0.961(0.857, 1.078)	0.964(0.829, 1.122)	**1.222(1.022, 1.461)**
*p* for trend	0.511	0.536	0.514	0.153	0.474	0.413
NP4						
Q1	1.000	1.000	1.000	1.000	1.000	1.000
Q2	1.007(0.801, 1.266)	0.895(0.779, 1.029)	1.110(0.735, 1.677)	1.066(0.962, 1.182)	1.000(0.872, 1.147)	0.937(0.797, 1.100)
Q3	0.926(0.732, 1.172)	0.914(0.794, 1.051)	1.175(0.776, 1.779)	0.974(0.878, 1.082)	0.966(0.840, 1.109)	0.975(0.829, 1.147)
Q4	1.229(0.987, 1.529)	1.041(0.910, 1.190)	1.201(0.800, 1.804)	1.046(0.944, 1.159)	1.070(0.935, 1.224)	0.908(0.771, 1.070)
*p* for trend	0.419	0.625	0.359	0.579	0.708	0.331
NP5						
Q1	1.000	1.000	1.000	1.000	1.000	1.000
Q2	1.065(0.855, 1.327)	0.948(0.825, 1.089)	0.822(0.542, 1.246)	0.957(0.865, 1.060)	0.953(0.834, 1.089)	0.887(0.760, 1.036)
Q3	0.995(0.797, 1.243)	1.033(0.901, 1.184)	0.914(0.610, 1.369)	0.975(0.881, 1.080)	0.964(0.843, 1.102)	**0.763(0.650, 0.895)**
Q4	**0.782(0.623, 0.983)**	0.969(0.847, 1.109)	1.043(0.710, 1.532)	0.918(0.829, 1.015)	**0.839(0.733, 0.960)**	**0.650(0.552, 0.766)**
*p* for trend	0.235	0.813	0.805	0.177	0.058	**<0.0001**

The statistically significant results are shown in bold. The model was adjusted for sex, living area, age, energy intake, Engel’s coefficient, physical activity, passive smoking, alcohol drinking, and family size. ORs: odds ratios, 95%CIs: 95% confidence intervals.

## Data Availability

The data are not available according to the policy of the National Institute for Nutrition and Health, Chinese Center for Disease Control and Prevention.

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
