# Peer review of "Nutrient Patterns and Its Association and Metabolic Syndrome among Chinese Children and Adolescents Aged 7–17"

_nutrients, 2022, doi:10.3390/nu15010117_

Round 1

Reviewer 1 Report

Dear Editor,

I would like to congratulate the authors for this study in which they evaluated the associations between nutrient patterns and MetS and its components among Chinese children and adolescents aged 7-17 between 2016-2017.

Without a single definition, several definitions of MetS in children and adolescent are used. What was the reason why you used the Cook classification and not the IDF criteria?

Have you used age- and sex-specific percentiles for evaluating growth, weight, and abdominal obesity, dyslipidemia, elevated BP, and impaired glucose metabolism in Chinese children and adolescents?

Were patients with secondary obesity (e.g chronic corticosteroid therapy, chemotherapy) excluded from the study?

Thank you!

Author Response

Dear reviewer,

Thank you for your letter and comments concerning our manuscript entitled "Nutrient Patterns and its Association and Metabolic Syndrome Among Chinese Children and Adolescents Aged 7-17"(nutrients-2112846). It’s very glad to have your recognition of our work. In the future studies, we will maintain our focus on the diet and health  of Chinese children and adolescents, and strive to contribute to the healthy growth and development of them.

Warm regards,

Reviewer 2 Report

I would like to thank you for giving me this opportunity to evaluate this scientific paper, which focuses on the associations between nutrient patterns (NPs) and 15 metabolic syndrome and its components among Chinese children and adolescents aged 7- 17.

This manuscript reports new findings and is theoretically based on the current literature.

The subject is very important for nutrition research.

- The manuscript is within the journal's scope.

- The sample sized of 13,071 children and adolescents is very impressive

- This study was well designed, executed, and presented.

- Figures and tables are well presented

- The conclusion is consistent with the evidence presented

- The discussion is relevant

- References are up to date and relevant

In Material and Methods please describe the ethics regarding this study, as the subjects are children, vulnerable population in research please give a few details about ethics approval and informed consent.

For the table 1 and 2 where you have the Nutrient patterns (NP1…5) add a name for each NP.

Author Response

Dear reviewer,

Thank you for your letter and comments concerning our manuscript entitled "Nutrient Patterns and its Association and Metabolic Syndrome Among Chinese Children and Adolescents Aged 7-17"(nutrients-2112846). It’s very glad to have your recognition of our work. Based on your suggestions, we have accordingly revised our manuscript.  We hope that our answers have satisfied your comments and look forward to your response.

Warm regards,
